# Hidden Dangers: Herbal and Dietary Supplement Induced Hepatotoxicity

Jonathan Kwong-Shing Lin  and Shannan R. Tujios *

Division of Digestive and Liver Disease, University of Texas Southwestern Medical Center, Dallas, TX 75390-8887, USA
* Correspondence: shannan.tujios@utsouthwestern.edu

**Abstract:** Herbal and dietary supplements represent a multi-billion-dollar industry reportedly used by over half of American adults. However, these products are not regulated by the Federal Drug Agency and contain a wide range of contaminants, leading to over 50,000 adverse events each year. This review aims to highlight the widespread use and current regulatory status of herbal and dietary supplements, identify the presentation and diagnostic dilemmas faced with liver injury, and discuss the most common agents implicated in herbal and dietary supplement hepatotoxicity.

**Keywords:** drug-induced liver injury; herbal and dietary supplement; hepatotoxicity; complementary and alternative medicine



## 1. History of Herbal Medicine

While pharmacologic agents were not available until the 19th century, people have been using herbal products to treat illness for centuries. The oldest written evidence was found on a Sumerian clay slab over 5000 years old. In China, the discovery of herbal medicine has been ascribed to emperor Shen Nung (2696 BC). Descriptions of over 365 herbal remedies, including some still used today, were transcribed 2000 years later in the great herbal book, *Shen Nung Pen Tsao Ching*. Numerous herbs including turmeric were mentioned in Aryuvedic books in India nearly 3000 years ago. In Egypt, the Ebers Papyrus (ca. 1550 B.C.) details over 800 formulas of 700 different plants. The ancient Greeks and Romans were herbalists with Dioscorides compiling his *De Materia Medica* ca. 70 A.D. In this five-volume work, over 900 drugs are described from over 600 plant species. It was translated into many languages and served as the primary pharmacopeia for over 1500 years. The native people of Africa, the South Pacific Islands, and the Americas also had plant-centered healing traditions. Traditionally herbal products made of stems, roots, bark, leaves, berries, seeds, and flowers were administered as poultices, compresses, infusions, or decoctions.

In the early 1800s, the discovery and isolation of alkaloids from poppy for opiates (1806) and quinine from the bark of the *Cinchona* tree to treat malaria (1820) marked the beginning of modern pharmacology [1]. The value of botanicals to modern medicine is undeniable. Aspirin is derived from willow bark, digitalis from purple foxglove, vincristine from Madagascar periwinkle, and many others. It is estimated that 25 to 40% of modern prescription drugs contain at least one compound now or once derived from or patterned after those found in plants. Nearly 70% of antimicrobials and 50% of anticancer drugs developed in the last 30 years are derived from natural products [2]. However, reports of liver toxicity due to herbal and dietary supplements have been increasing worldwide over recent years signaling an important change from historical use.

## 2. Herbal and Dietary Supplements Today

More recently, there has been a growing interest in herbal and dietary supplements used as complementary and alternative medicine (CAM) which has spawned a multi-billion dollar industry. In a 1990 telephone survey, 34% of adult Americans reported

using CAM with 2.5% using herbals. This increased to 42% using CAM and 12.1% using herbals when the survey was repeated in 1997 [3]. The National Health Interview Survey reports 33% of adults using CAM 2002–2012, with non-vitamin dietary supplements most used at 18%. Based on the most recent 2012 survey, an estimated 59 million Americans spend USD 30.2 billion a year on CAM with USD 12.8 billion on natural products [4]. If vitamin preparations are included, over half of U.S. adults report using supplements fueling the USD 37.6 billion a year dietary supplement market. Users are more likely to be non-Hispanic white females over the age of 40 with higher levels of income and education. Most report using supplements for wellness but 40% do so to treat a condition. Because these products are readily available and "natural", they are perceived to be effective and safe. However, herbal, and dietary supplements (HDS) are not classified as drugs and, by definition, are not intended to prevent, diagnose, treat, mitigate, or cure diseases [5]. Citing lack of questioning to fear of disapproval from their primary care physicians, over 40% do not disclose the use of CAM, including a quarter of those taking supplements [6].

### 3. Regulation of Herbal and Dietary Supplements

In order to ensure the American public had access to safe foods and drugs, the Pure Food and Drug Act was passed in 1906 heralding the beginning of the modern-day Food and Drug Administration (FDA). In subsequent years further regulations were passed in response to deaths due to poorly manufactured drugs. In 1938, the Food, Drug and Cosmetic Act required premarketing safety data, and in 1962, amendments mandated that a drug's efficacy and safety be proved in clinical trials before approval. During this time, vitamins and minerals were considered over-the-counter drugs and regulated as such while herbals were considered foodstuffs.

In 1994, Congress passed the Dietary Supplement Health Education Act (DSHEA). It defines a supplement as a vitamin, mineral, herb, or other botanical, amino acid, or dietary substance available as a concentrate, metabolite, constituent, extract, or any combination meant to supplement the diet. This law now allows dietary supplements to be marketed without FDA approval. Unlike pharmaceutical companies, manufacturers of HDS need not prove safety and efficacy before marketing. Under the Good Manufacturing Practices of 2007, manufacturers are expected to provide truthful labeling, maintain quality standards, and ensure the safety of their products (Table 1) [7,8]. Adverse events can be reported by the manufacturer or through MedWatch (the FDA's medical product safety reporting program for healthcare professionals, patients, and consumers), at which point the FDA can make recommendations on removing a product from the market if necessary [9].

Despite these regulations, there is a wide range of variability among products and numerous reports of contaminants and adulterants. In an analysis of 78 bottles of herbal supplements, 80% contained no DNA from the plant advertised on the label. Another study found that 59% contained additional botanicals not listed on the label, including potential allergens and toxins. More than 500 "all-natural" products have been found to contain pharmaceuticals, especially in the case of those marketed for weight loss (sibutramine), sexual enhancement (phosphodiesterase-5 inhibitors), and athletic performance (amphetamines and anabolic steroids). In a recent 2019 paper, DILIN demonstrated that of the 272 HDS they assessed via high-performance liquid chromatography, 51% were mislabeled [10]. The FDA estimates 50,000 adverse events annually from HDS with most being related to renal and liver injury [11].

**Table 1.** Regulation of herbal and dietary supplements in the United States [8].

| Regulation | Responsibilities | |
|---|---|---|
| | **Manufacturer** | **FDA** |
| DSHEA (1994) | • Identify product ingredients and manufacturer on the label<br>• Provide disclaimer noting that product was not evaluated by the FDA for safety and efficacy, and is not intended to diagnose, treat, cure, or prevent disease | • Defines supplements as vitamins, minerals, herbs, amino acids (and any concentrate, metabolite, extract thereof)<br>• Investigate allegations of attributable toxicity after marketing<br>• Conducts premarket review of safety data for new ingredients |
| cGMP (2007) | • Must adhere to standards in identification, purity, strength, composition, and purity of the final dietary supplement<br>• Must evaluate the identity, purity, strength, and composition of dietary supplements | • Supplements containing contaminants or not containing labeled ingredients are considered adulterated or misbranded |

Abbreviation: cGMP, current good manufacturing practice.

## 4. Incidence of Herbal and Dietary Supplement-Induced Liver Injury

The true incidence of herbal hepatotoxicity is unknown and thought to be underreported. It is estimated that less than 1% of adverse reactions to HDS are reported [12]. As with drug-induced liver injury (DILI) from conventional pharmaceuticals, most cases of HDS-induced liver injury are idiosyncratic rather than predictable, dose-dependent reactions as seen with acetaminophen. Information is gleaned from case reports, retrospective reviews, and recent prospective databases. The Drug Induced Liver Injury Network (DILIN), a multicenter United States research collaboration, found the proportion of cases attributed to HDS increased from 7% in 2004 to 2005 to 20% from 2013 to 2014 [13]. Between 2004 to 2019, 369 cases of liver injury due to HDS were enrolled into DILIN—representing 19% of DILI cases [14]. Other Western registries report similar percentages while hepatotoxicity due to herbals contributed to over 70% of cases in Korea and Singapore and 40% in China [15].

Acute liver failure, defined as elevated liver enzymes with INR > 1.5 and encephalopathy, from prospective DILI studies occurred in 1.5 to 11% of HDS cases. The US Acute Liver Failure Study Group reported 253 (9.6%) patients with idiosyncratic DILI with 16.3% of those related to HDS between 1998 and 2015. Mirroring the DILIN's findings, the fraction of HDS among drug-induced acute liver failure increased over time from 12.4% from 1998–2007 up to 21.1% 2007–2015. Alarmingly, HDS-induced acute liver failure resulted in transplant or death more often than cases due to prescription medication (83% vs. 66%) [16]. In a 2015 population study, over 18% of acute liver failure cases were attributed to HDS with 50% resulting in death or liver transplant [17].

## 5. Diagnosing Herbal and Dietary Supplement-Induced Liver Injury

Most adverse drug and herbal reactions are idiosyncratic—due to an individual's own mix of unique characteristics—rather than a predictable occurrence. Proposed mechanisms of drug-induced hepatocyte injury include the formation of chemically reactive metabolites causing direct cell lysis, damage to bile salt export pumps, mitochondrial inhibition, stimulating apoptotic pathways, and formation of drug haptens that activate the immune response.

The diagnosis of HDS-induced liver injury follows the same principles as drug-induced liver injury but with some added challenges. Keys to diagnosing drug and HDS liver injury include:

- Exposure must precede the onset of liver injury (although the time from start to injury is highly variable);
- More common causes of liver disease should be excluded;
- Injury typically improves when the causative agent is stopped ("dechallenge");
- Liver injury may recur more rapidly and severely after repeated exposure ("rechallenge").

The clinical presentation of hepatotoxicity can vary, ranging from asymptomatic elevated liver tests to fatigue, nausea, and abdominal pain to jaundice and encephalopathy. Once liver injury is recognized and other causes of liver injury are excluded, there must be a high index of suspicion that it may be due to supplement use. Mitigating factors include the patient not divulging HDS, irregularities in HDS, and unfamiliarity by both the patient and clinician of potential culprits. While the timing of symptoms with the beginning of HDS is suggestive, products consumed previously must also be considered as contents may vary over time.

The pattern of liver injury provides a clue as many drugs and some HDS have characteristic signatures. The R ratio (ALT value/ALT ULN)/Alk P value/Alk P ULN) can be calculated to determine if hepatocellular (R ratio > 5), cholestatic (R ratio < 2), or mixed (R ratio 2–5). Clinical improvement after HDS is stopped supports diagnosis but some go on to have progressive liver injury. Incidental re-exposure with the return of symptoms and liver test abnormalities strengthens diagnosis as well (Figure 1) [18].

### 5.1. Determining Severity of Liver Injury

#### 5.1.1. Hy's Law

In 1978, Dr. Hyman Zimmerman observed that "drug-induced hepatocellular jaundice is a serious lesion" resulting in 10–50% mortality. This observation has been validated throughout the years and is known as "Hy's Law." When evaluating new drugs, the FDA uses Hy's law (aminotransferases > 3 × ULN with bilirubin > 2 × ULN with normal alkaline phosphatase probably due to drug = 1/10 death) to identify signals for severe hepatotoxicity that may occur post-marketing. If 1 Hy's law case is identified in 1000, this suggests severe liver injury at a rate of 1/10,000. It is worrisome to find one Hy's Law case in a clinical trial database but finding two is considered highly predictive that the drug has the potential to cause severe liver injury when given to a larger population [19]. It is important to realize that the severity of liver injury is not related to the degree of transaminase elevation but to the presence of hyperbilirubinemia and/or prolonged prothrombin time indicating hepatic dysfunction.

#### 5.1.2. Assigning Causality

Due to the lack of an objective test, attributing liver injury to a drug or HDS can be difficult. DILI can mimic almost any other known liver disease, has differing histologic findings, and overall is a rare occurrence. DILI is a diagnosis of exclusion based on clinical assessment and ruling out other potential etiologies with laboratory testing. Several scoring systems have been developed for DILI and extrapolated for use in HDS-induced liver injury. The Council for International Organizations of Medical Sciences (CIOMS) created the Roussel Uclaf Causality Assessment Method (RUCAM) in 1989 as the first liver-specific causality tool. This score assigns a numerical value to 7 different factors (chronology, risk factors, concomitant drug use, exclusion of other causes, reported history of drug's toxicity, and response to dechallenge) with scores −8 to 14 to grade probability as definite, very likely, probable, possible, unlikely, or excluded (Figure 2). In validation studies, RUCAM had 93% positive predictive value and 78% negative predictive value. However, criticism of this method includes weight placed on age, pregnancy, history of hepatotoxicity, and unknown alcohol amount as a cofactor.

**Algorithm for the assessment of suspected HILI**

*Hepatitis A, B, C, E, CMB, EBV, HSV, VZV, autoimmune hepatitis, alcoholic liver disease, ischemic liver injury/hemodynamic collapse, genetic liver diseases, biliary obstruction, vascular injury

R ratio = (ALT value/ALT ULN)/(Alk P value/Alp P ULN)

**Figure 1.** From Rossi et al. [18].

The DILIN group uses a structured three-expert opinion process to assign causality based on likelihood percentage. Cases are assigned to one of five categories ranging from unlikely to definite with severity ranging from mild to fatal. The DILIN method achieves high inter-observer agreement and is more likely to diagnose DILI when compared to RUCAM. However, the DILIN consensus method is cumbersome, not available to the general clinician, and subjective (Table 2). Despite best efforts to objectively diagnose liver injury due to drug and HDS, adjudication remains more of an art rather than science [20].

| RUCAM   Causality Assessment | | | | | |
|---|---|---|---|---|---|
| Drug: _______________________________     Initial ALT: __________     Initial Alk P: __________     R ratio = [ALT/ULN] ÷ [Alk P/ULN] = _______ ÷ ________ = ________ | | | | | |
| The R ratio determines whether the injury is hepatocellular (R > 5.0), cholestatic (R < 2.0), or mixed (R = 2.0 – 5.0) | | | | | |
| | Hepatocellular Type | | Cholestatic or Mixed Type | | Assessment |
| **1. Time to onset** | | | | | |
| | **Initial Treatment** | **Subsequent Treatment** | **Initial Treatment** | **Subsequent Treatment** | **Score** (check one only) |
| o   From the beginning of the drug:<br>  • Suggestive<br>  • Compatible | 5 – 90 days<br>< 5 or > 90 days | 1 – 15 days<br>> 15 days | 5 – 90 days<br>< 5 or > 90 days | 1 – 90 days<br>> 90 days | ☐ +2<br>☐ +1 |
| o   From cessation of the drug:<br>  • Compatible | ≤ 15 days | ≤ 15 days | ≤ 30 days | ≤ 30 days | ☐ +1 |
| **Note:** If reaction begins before starting the medication or >15 days after stopping (hepatocellular), or >30 days after stopping (cholestatic), the injury should be considered unrelated and the RUCAM cannot be calculated. | | | | | |
| **2. Course** | **Change in ALT between peak value and ULN** | | **Change in Alk P (or total bilirubin) between peak value and ULN** | | **Score** (check one only) |
| **After stopping the drug:** | | | | | |
| • Highly suggestive | Decrease ≥ 50% within 8 days | | Not applicable | | ☐ +3 |
| • Suggestive | Decrease ≥ 50% within 30 days | | Decrease ≥ 50% within 180 days | | ☐ +2 |
| • Compatible | Not applicable | | Decrease < 50% within 180 days | | ☐ +1 |
| • Inconclusive | No information or decrease ≥ 50% after 30 days | | Persistence or increase or no  information | | ☐ 0 |
| • Against the role of the drug | Decrease < 50% after 30 days **OR**<br>Recurrent increase | | Not applicable | | ☐ -2 |
| o   **If the drug is continued:**<br>  • Inconclusive | All situations | | All situations | | ☐ 0 |
| **3. Risk Factors:** | **Ethanol** | | **Ethanol or Pregnancy (either)** | | **Score**<br>(check one for each) |
| o   Alcohol or Pregnancy | Presence<br>Absence | | Presence<br>Absence | | ☐ +1<br>☐ 0 |
| o   Age | Age of the patient ≥ 55 years<br>Age of the patient < 55 years | | Age of the patient ≥ 55 years<br>Age of the patient < 55 years | | ☐ +1<br>☐ 0 |

**Figure 2.** *Cont.*

| 4. Concomitant drug(s): | | | Score (check one only) |
|---|---|---|---|
| o None or no information or concomitant drug with incompatible time to onset | | | ☐ 0 |
| o Concomitant drug with suggestive or compatible time to onset | | | ☐ -1 |
| o Concomitant drug known to be hepatotoxic with a suggestive time to onset | | | ☐ -2 |
| o Concomitant drug with clear evidence for its role (positive rechallenge or clear link to injury and typical signature) | | | ☐ -3 |
| **5. Exclusion of other causes of liver injury:** | | | **Score** (check one only) |
| **Group I (6 causes):**<br>o **Acute viral hepatitis** due to **HAV** (IgM anti-HAV), or<br>o  **HBV** (HBsAg and/or IgM anti-HBc), or<br>o  **HCV** (anti HCV and/or HCV RNA with appropriate clinical history)<br>o **Biliary obstruction** (By imaging)<br>o **Alcoholism** (History of excessive intake and AST/ALT ≥ 2)<br>o **Recent history of hypotension, shock or ischemia** (within 2 weeks of onset)<br>**Group II (2 categories of causes):**<br>o Complications of underlying disease(s) such as autoimmune hepatitis, sepsis, chronic hepatitis B or C, primary biliary cirrhosis or sclerosing cholangitis; or<br>o Clinical features or serologic and virologic tests indicating acute CMV, EBV, or HSV. | o All causes in Group I and II ruled out | | ☐ +2 |
| | o The 6 causes of Group I ruled out | | ☐ +1 |
| | o Five or 4 causes of Group I ruled out | | ☐ 0 |
| | o Less than 4 causes of Group 1 ruled out | | ☐ -2 |
| | o Non drug cause highly probable | | ☐ -3 |
| **6. Previous information on hepatotoxicity of the drug:** | | | **Score** (check one only) |
| o Reaction labeled in the product characteristics | | | ☐ +2 |
| o Reaction published but unlabeled | | | ☐ +1 |
| o Reaction unknown | | | ☐ 0 |
| **7. Response to readministration:** | | | **Score** (check one only) |
| o Positive | Doubling of ALT with drug alone | Doubling of Alk P (or bilirubin) with drug alone | ☐ +3 |
| o Compatible | Doubling of the ALT with the suspect drug combined with another drug which had been given at the time of onset of the initial injury | Doubling of the Alk P (or bilirubin) with the suspect drug combined with another drug which had been given at the time of onset of the initial injury | ☐ +1 |
| o Negative | Increase of ALT but less than ULN with drug alone | Increase of Alk P (or bilirubin) but less than ULN with drug alone | ☐ -2 |
| o Not done or not interpretable | Other situations | Other situations | ☐ 0 |
| | | **TOTAL (add the checked figures)** | |

*Abbreviations used: ALT, alanine aminotransferase; Alk P, alkaline phosphatase; ULN, upper limit of the normal range of values*
*Modified from: Danan G and Benichou C. J Clin Epidemiol 1993; 46: 1323-30.*

**Figure 2.** RUCAM Causality Assessment Available online at: https://www.ncbi.nlm.nih.gov/books/NBK548272/bin/livertoxrucamv5.pdf (accessed on 30 August 2023).

**Table 2.** Comparison of RUCAM, clinical practice expert opinion, and DILIN expert consensus [20].

| Criteria | RUCAM | Clinical Practice | DILIN |
|---|---|---|---|
| Adjudication process | Semi-objective quantitative scoring method | Individual expert opinion | Expert consensus |
| Clinical Setting | Real-time, retrospective | Real-time | Case reviewed within 6 months of acute onset |
| Reviewers | 1 | 1 | 3 |
| Categories | Highly probable, probable, possible, unlikely, excluded | Likelihood of 50% or higher to support clinical decision | Definite, highly likely, probable, possible, unlikely |
| Duration of Follow up | 1–3 months | Days to months | 6 months and longer |
| Reliance on a positive response to rechallenge | Strong | Rarely | Rarely |
| Ease of use | Not formally tested | Dependent on clinical experience | Limited to DILIN experts |
| Reproducibility | Kappa = 0.34 May improve with increasing use and familiarity | Unknown, improves with discussions amongst experts arriving at a consensus | Kappa = 0.6 Relies on discussion to reach consensus when individual assessments differ |

### 5.1.3. Herbal and Dietary Supplements Linked to Liver Injury

Determining the agent(s) responsible for HDS-induced liver injury is uniquely challenging. The permissive regulatory conditions surrounding HDS allow the potential for direct herbal hepatotoxins, contaminants, and adulterants to be released on the market without any oversight, determination of safety, or efficacy at any stage. To further complicate matters, most cases of HDS-induced liver injury implicate commercial products containing numerous ingredients. Finally, products vary often over time in the list of ingredients contained with no surveillance required. To recognize potential culprits, researchers have categorized products by marketed intent. In the DILIN, the most reported products were those used for bodybuilding, but the non-bodybuilding HDS tended to cause a more severe liver injury (Figure 3) [13].

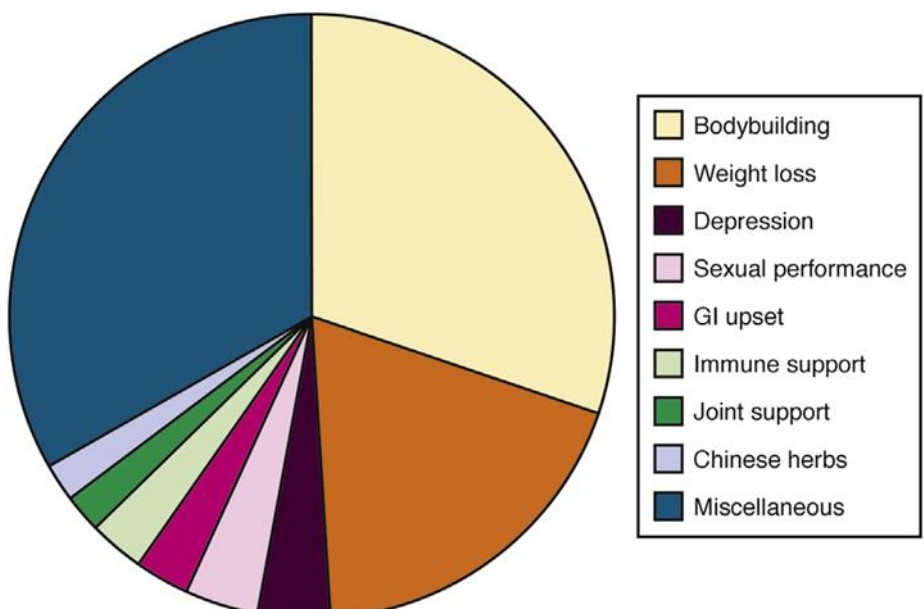

**Figure 3.** Marketed intent of HDS with liver injury [13].

*5.2. Bodybuilding Supplements*

The number of cases of DILI from bodybuilding HDS has been on the rise with bodybuilding HDS making up 2% of DILIN cases from 2004–2005 to 8% of cases from 2010 to 2012. The affected population is more likely to be young males (median age 31) [21]. The clinical scenario and pattern of liver injury seen with bodybuilding supplements are virtually diagnostic of anabolic androgenic steroids. Anabolic androgenic steroids are synthetic derivatives of testosterone with restricted use since 1991 due to abuse potential. Despite being labeled as Class III controlled substances, they remain widely available through the Internet as dietary supplements. To circumvent legislation banning these substances, new forms of anabolic steroids are being synthesized making detection of these "designer steroids" more difficult. One study found that 23 out of 24 bodybuilding products were adulterated with anabolic steroid compounds, most with no mention on the packaging [7].

Anabolic androgenic steroids cause several types of liver injury. Focal nodular hyperplasia, peliosis hepatitis, adenomas, hepatocellular carcinoma, and even steatosis have all been described though intrahepatic cholestasis is classic and the most frequent form. Typical cases are young men who present with jaundice, pruritus, and weight loss, having purchased the product at their gym. Liver enzymes are usually cholestatic though hepatocellular cases have been reported. Most recover but have prolonged jaundice lasting 3–12 months. The exact mechanism of injury is unknown, but animal studies showed anabolic steroid-induced oxidative stress impaired the bile salt export pump [22]. Given known mutations in ATP8B1/ABCB11, bile homeostasis, and export proteins, genetic susceptibility for steroid-induced cholestasis may exist, although one has not been identified to date [23].

*5.3. Weight Loss Supplements*

5.3.1. Hydoxycut®

Hydroxycut® (Iovate Health Sciences, Oakville, ON, Canada), a multi-ingredient nutritional supplement purported to be "America's #1 selling weight loss supplement brand," is a family of dietary supplements marketed for weight loss. Initial products released in 2002 contained Ma Huang, a botanical containing ephedra alkaloids, and was linked to 2 cases of acute hepatocellular injury with jaundice. After reports of cardiovascular, neurological, and hepatic compromise, the FDA banned dietary supplements containing ephedra in 2004. Despite removing ephedra, accounts of liver injury associated with Hydroxycut® continued with nine additional cases reported. A recent series describes an additional 17 cases of Hydroxycut® associated severe liver injury. Nine men and eight women were young (mean age 30 years), symptomatic, and presented with hepatocellular injury with transaminases over 1000 U/L and jaundice an average of 6.4 weeks after starting the supplement. Positive antinuclear antibodies were not uncommon, and histology showed cholestatic hepatitis or massive necrosis. In contrast to previous cases with spontaneous recovery, all required hospitalization, three underwent liver transplants and one died. The specific ingredient or combination of ingredients in Hydroxycut® responsible for hepatoxicity is unclear. *Cissus quadrangularis*, *Gymnema sylvestre*, *Garcinia cambogia*, *Camellia sinensis* (green tea extract), chromium, and caffeine have been suggested as potential hepatotoxins but most speculate that liver injury is due to green tea extract [24].

In 2009, the FDA issued a public warning regarding the risk of severe liver injury associated with Hydroxycut® and the manufacturer recalled its products. Within a month of the recall, new Hydroxycut® formulations containing none of the original botanicals were released. While most patients with Hydroxycut® associated liver injury recover without long-term sequelae, new cases of liver injury are already being implicated [7,25] and a case of vanishing bile duct syndrome has been published recently [26].

### 5.3.2. Herbalife[®]

Herbalife[®] (Los Angeles, CA, USA) is one the largest weight management and supplement companies in the world with sales of over USD 5 billion. Products in the form of drinks, energy bars, tablets, and capsules are exported to over 60 countries. Since 2005 there have been over 60 case reports of liver injury from countries including Switzerland, Israel, Spain, Argentina, Iceland, USA, and Venezuela [27]. Most cases are hepatocellular but mixed and cholestatic patterns of liver injury have also been observed. Using various causality scores, most cases were considered probable though definite cases with positive re-challenge were reported. Severity ranges from mild to severe with cases of acute liver failure requiring transplant and chronic injury with eventual cirrhosis reported. Due to the multitude of products containing numerous ingredients, including green tea extract, identifying a potential toxin and mechanism is difficult [28]. Two Herbalife[®] cases identified *Bacillus subtilis* as a contaminant with dose-dependent leakage of lactate dehydrogenase (LDH) from hepatocytes [29]. However, the company maintains it follows strict quality control and challenges the validity of these reports.

### 5.3.3. Green Tea Extract (*Camellia sinensis*)

Tea, the most widely consumed beverage next to water, is prepared by pouring hot water over the leaves of the evergreen shrub *Camellia sinensis*. The major bioactive compounds in tea are polyphenol flavonoids called catechins with epigallocatechin gallate (EGCG) making up 30–50% of the dry weight of green tea. The average cup of green tea contains 50–150 mg of EGCG. However, concentrated green tea extracts are now a common ingredient in HDS, especially those promoting weight loss [30].

Since 1999, numerous studies involving at least 90 cases have implicated products allegedly containing green tea extract in liver injury [27]. In 2003 Exolise[®], an alcoholic extract of *Camellia sinensis*, was withdrawn in France and Spain after 13 cases of acute liver damage. Most reported cases present with hepatocellular injury patterns within 3 months of start and recover after cessation. At least seven patients exhibited accelerated recurrence of liver injury after rechallenging [31]. The US Pharmacopeia reviewed the 34 published cases through 2008 and concluded while there was cause for concern, no advisory label was required [32]. An additional 19 cases of liver injury associated with green tea extract have been identified from 2008 to 2015 with two-thirds involving products with multiple ingredients [33]. Additionally, among the 97 HDS taken by 47 subjects in the DILIN, 51% (49/97) contained green tea extracts with 40% (29/73) not indicating its presence on the product label. However, there was no statistically significant association between the catechin presence and liver injury causality score, severity, or pattern of liver injury. Catechin is the component of green tea implicated in liver injury, and catechin levels tend to be highest in products used for weight loss but levels are highly variable [34].

The mechanism of green tea extract-associated liver injury is unknown but thought to be due to catechin. EGCG has been shown to be a dose-dependent hepatotoxin in mice causing hepatic necrosis with oxidative stress. It is purported that EGCG may induce reactive oxygen species and affect the mitochondrial membrane in certain conditions, such as fasting or genetic predisposition [35]. In a 2021 study, researchers have also discovered a close association between green tea-related liver injury and the allele HLA-B*35:01 which is present in 5–15% of the American population. From 2004–2018, 38 unique patients in the DILIN were judged to have definite, highly likely, or probable green tea-related liver injury, and 26 of these patients had at least one copy of the HLA-B*35:01 allele, yielding a carrier frequency of 72%. The results of the study demonstrate that in green-tea-associated liver injury, there can be an idiosyncratic and immune-mediated component [36].

In contrast to hepatotoxicity concerns, experimental and clinical data support the potential benefits of green tea extract. Furthermore, the overall occurrence of liver injury related to green tea appears low. In a meta-analysis of 4 randomized controlled trials of green tea extract that reported adverse liver events, 8 occurred among 1405 subjects (0.5%) versus 1 in 1200 controls (0.1%). Mildly elevated liver enzymes were seen at daily doses

800–1600 mg EGCG with one severely elevated ALT resulting in discontinuation of 1600 mg dose. No serious events occurred [37]. The safety of green tea extract is likely due to methods of extraction and preparation, interaction between ingredients, as well as an individual's genetic background.

### 5.4. Usnic Acid

Usnic acid, derived from lichens, is an uncoupler of oxidative phosphorylation in mitochondria. It is believed that decreased efficacy of energy use may lead to thermogenesis and increased fat metabolism for weight loss. However, this mechanism leads to increased oxidative stress, free radical generation, and hepatocyte death. Several cases of acute hepatocellular liver injury have been attributed to HDS containing usnic acid, including ones resulting in liver transplant and death. *LipoKinetix®*, a supplement advertised for weight loss which contained 100 mg of sodium usniate, norephedrine, diiothyronine, yohimbe, and caffeine and *UCP-1®* which contained 150 mg of usinic acid, L-carnitine, and calcium pyruvate together were responsible for at least 21 cases of liver injury. Of these cases, seven developed acute liver injury, one underwent a liver transplant, and one died. Given these findings, the FDA issued a warning in 2001 and both LipoKinetix® and UCP-1 were removed from the market that same year [38].

### 5.5. OxyELITE Pro™

On September 9, 2013, the Hawaii Department of Health was notified of seven previously healthy adults presenting from May to September 2013 with severe acute hepatitis and liver failure of unknown etiology. All were reported using OxyELITE Pro™ (USPLabs, Dallas, TX, USA), a multi-ingredient supplement containing "proprietary blends of plant-derived extracts" marketed for weight loss [39]. The Hawaii Department of Health with the Centers for Disease Control and the FDA initiated a public health investigation requesting any cases of liver injury associated with weight loss or muscle-building supplements. Clinicians submitted 76 reports, 44 of which fulfilled the criteria of HDS liver injury, with a total of 36 individuals identified in Hawaii with suspected OxyELITE Pro™ associated liver injury. Nearly two-thirds reported using supplements in addition to OxyELITE Pro™ and time from initial use to the onset of symptoms ranged from 1 week to 2 years. Subjects were young (median age 33 years), 57% female, with hepatocellular injury pattern (medians of ALT, AP, and Tbili 1740 IU/L, 141 IU/L, and 9.4 mg/dL). Fourteen patients required hospitalization, 2 received liver transplants and 1 died of cerebral edema.

In early 2013, USPLabs reformulated OxyELITE Pro™ due to the FDA's ban on 1,3-dimethylamylamine (DMAA), an ingredient associated with acute myocardial infarction. A review of the liver injury cases revealed that most reported taking the new DMAA-free formulation of "Super Thermo" OxyELITE Pro™ in the weeks prior to symptoms. Of note, the new formulation contained aegeline, a compound from the native Ayurvedic herbal *Aele marmelos*. The aegeline ingredient, synthesized in China, was a new dietary ingredient and contrary to DSHEA regulations, had not been disclosed to the FDA prior to marketing. No additional known hepatoxins, contaminants, or adulterants were found in the products analyzed by the FDA [40]. While all other etiologies of acute liver injury were reportedly excluded, a recent review criticized the causality assessments of the Hawaii cases and concluded that there was insufficient evidence to incriminate OxyELITE Pro™ in all reports [41]. However, an additional seven cases with similar characteristics presenting in 2013 were identified in the continental US through the DILIN, three with acute liver failure, and two requiring liver transplant [42]. The FDA issued a warning letter and OxyELITE Pro™ formulations containing aegeline were removed in November 2013.

Additional HDS marketed for weight loss that have been linked to liver injury include linoleic acid, Ma huang (*Ephedra sinica*), *Garcinia cambogia*, Germander (*Teucrium chamaedrys*), Chaparral (*Larrea tridentata*), herbals "Onshidou-Genbi-Kounou" and "Chaso" [43]. At least 10 cases of hepatocellular liver injury including acute liver failure have been linked to *Ephedra.* Due to its association with cardiovascular events and stroke, the FDA banned

products containing *Ephedra* in 2004 [31]. The popular *Garcinia cambogia* has been implicated in at least a dozen cases of liver injury including acute liver failure [28,44].

### 5.6. Health Supplements
Turmeric

Turmeric, an herbal supplement with reported anti-inflammatory properties, has seen a significant rise in popularity during the COVID-19 pandemic when it was advertised as a common preventative remedy. Turmeric is derived from the roots of the plant *Curcuma longa* which belongs to the ginger family. The responsible anti-inflammatory component of turmeric is thought to be from curcuminoids such as curcumin. Turmeric, which is also a staple spice in South Asian cuisine, was previously thought to have a relatively safe hepatic profile, especially given curcumin's poor bioavailability in its oral form. However, from 2011–2022, ten cases of turmeric-associated liver injury have been enrolled in the DILIN network, with six of the ten cases reported since 2017. Of these ten patients, one died from acute liver failure. One reason for the rise in turmeric-associated liver injury is thought to be due to the development of new turmeric supplements using piperine (black pepper) and nanoparticle delivery methods which significantly increases the bioavailability of curcumin [45]. The onset of liver injury, if it transpires, often occurs within one to three months of initiating the supplement. Most patients develop a hepatocellular pattern of liver injury with transaminases that can rise to levels above 1000 U/L. In some cases, liver biopsies can reveal an autoimmune pattern of liver injury, prompting clinicians to initiate steroids, though the exact benefit remains unknown given liver injury often rapidly resolves once the offending agent is stopped [46].

### 5.7. Black Cohosh (Actaea racemosa, Syn Cimicufuga racemosa)

Black cohosh is a popular herbal product used primarily for menopausal symptoms. Since 2012, it has been in the top 5 selling botanicals in the US with over USD 40 million in annual sales. It is derived from the roots of *Actaea racemosa*, a plant native to North America. Over 50 reports of hepatotoxicity ranging from elevated transaminases, and autoimmune-like hepatitis to acute liver failure prompted a review by the US Pharmacopeia and a cautionary statement added to the label [47]. However, black cohosh has not been associated with liver injury in clinical trials involving over 1200 patients. Several cases of black cohosh hepatotoxicity have been attributed to adulterated products [48]. Driven by high demand, over-harvesting of native habitats, and increasing costs of authentic black cohosh, Asian *Actaea* are being intentionally substituted for economic gain. Analyses have shown that as many as 25% of preparations labeled as black cohosh contain Asian *Actaea* species instead [49]. US sellers through the Internet have the highest rate of adulterated products (38%) while no adulterated black cohosh preparations were discovered in those obtained from European pharmacies. To further complicate matters, there are now reports of adulterants of the Asian *Actaea* adulterants (www.botanicaladulterants.org, accessed on 30 June 2016). While black cohosh appears to be safe, patients may be at risk of liver injury from toxic products erroneously labeled as black cohosh.

### 5.8. Kava kava (Piper methysticum)

Traditionally a ceremonial beverage in the South Pacific Islands, kava is now sold in a variety of forms as an anxiolytic supplement. Nearly 100 reports worldwide of kava-associated liver injury including death led to its removal in some European markets and an FDA advisory in 2002. The mechanism of hepatoxicity is unknown. It has been postulated that toxicity may be due to alcoholic extraction leading to concentrated kava lactones, increased susceptibility in poor metabolizers with polymorphisms of CYP2D6, or contamination by mold, bacteria, or leaf alkaloids. Quality control and a daily dose of kava lactones not exceeding 250 mg have been proposed to decrease the risk of rare idiosyncratic liver injury [50]. Patients generally develop a hepatocellular pattern of liver injury within 6 months of initiating the supplement with some developing an immunoallergic presen-

tation including rash fever, and eosinophilia. In severe cases, liver biopsies can reveal hepatocellular necrosis and intrahepatic cholestasis [51].

### 5.9. Pyrrolizidine Alkaloids

One of the best-explained plant-associated liver injury comes from botanicals containing 1,2 dehydropyrrolizidine alkaloids. More than 350 pyrrolizidine alkaloids have been identified in over 6000 plant species with half of them thought to be hepatotoxic. The concentration of pyrrolizidine alkaloids may fluctuate according to the environment, climate, and season as well as the plant age and part. The potential for these agents to cause liver damage has been recognized for over 70 years. First described in Jamaican children drinking bush tea, hepatotoxicity has been reported with Crotalaria, Senecio, Heliotropium, and Symphytium (Comfrey) species.

Pyrrolizidine alkaloids are metabolized by cytochrome P450 into active pyrrole derivatives that then form cellular adducts (Figure 4). Inducers of P450 such as phenobarbital enhance toxicity while glutathione repleting N-acetyl-cysteine can be protective. Pyrrolizidine alkaloid adducts act as direct dose-dependent liver toxin damaging the sinusoidal epithelium with liver biopsy showing veno-occlusive disease. Presentation with hepatomegaly and ascites is pathognomonic. Mortality rates reach 20–40% with some developing cirrhosis. In 2001, the FDA banned oral Comfrey (Table 3) [15].

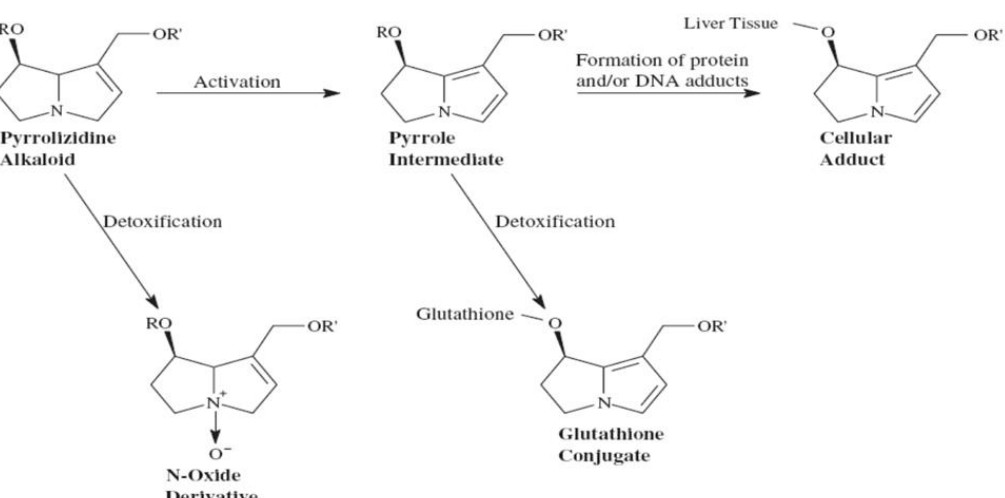

**Figure 4.** Mechanism of Comfrey hepatotoxicity [52].

**Table 3.** U.S. Food and Drug Administration warnings and recalls for liver injury associated with herbal and dietary supplements [43].

| Product | Year | Comments |
|---|---|---|
| Comfrey | 2001 | Contained pyrrolizidine alkaloids |
| Lipokinetix | 2001 | Contained phenylpropanolamine, caffeine, yohimbine, diiodothyronine, and sodium usniate |
| Kava | 2002 | Caused acute liver failure, hepatitis, cirrhosis |
| Hydroxycut | 2009 | Caused acute liver failure |
| Uprizing 2.0 | 2011 | Found to contain Superdrol (anabolic steroid) |
| OxyElite Pro | 2013 | Caused acute liver failure |

### 5.10. Vitamin A

It has been known for many years that high doses of vitamin A can result in liver damage. Toxicity is dose-dependent with excess vitamin A stored in the stellate cells. Excess accumulation of Vitamin A leads to stellate cell activations which in turn lead to cellular

hypertrophy, collagen deposition, and liver fibrosis [53]. A liver biopsy showing lipid droplet-laden stellate cells is diagnostic. Hepatotoxicity can present as elevated transaminases, cholestasis, non-cirrhotic portal hypertension, or cirrhosis several months to years after taking more than 10 times the recommended daily allowance. Over 50 hepatotoxicity cases due to vitamin-A-containing supplements have been published. While liver injury rarely occurs with an intake of less than 50,000 IU per day, there have been reports of injury with 25,000 IU per day and alcoholics may be more susceptible [28].

### 5.11. Kratrom

Derived from the leaves of the evergreen tree Mitragyna speciosa, kratom is often used for its purported ability to increase energy, and its ability to treat chronic pain, cough, and diarrhea. In recent years, kratom has gained increasing popularity for its ability to help with opioid withdrawal symptoms as well as its cannabis-like effects. However, its medical benefits have not been proven and overdoses have been associated with seizures, coma, and death [54]. In many countries, kratom has been labeled a drug of abuse rather than an herbal supplement, leading to its ban in countries such as Thailand and Malaysia. However, kratom is readily available for purchase in the United States via the Internet, though it is banned in certain states [55].

From 2003 to 2019, 11 cases in the DILIN were judged to have definite, highly likely, or probable kratom-related liver injury. The onset of symptoms occurred within a median of 14 days of using kratom, and all 11 patients developed jaundice with a peak total bilirubin ranging from 4.5–41.1 mg/dL. Most patients presented with a mixed pattern of liver injury, but cholestatic and hepatocellular patterns of liver injury were also possible. Liver enzymes completely normalized by 6 months to 1 year. None of these patients died from liver acute failure, however, the CDC has reported kratom-related deaths secondary to respiratory depression and coma [14]. Steroids and N-acetylcysteine have been intermittently used in cases of suspected kratom-induced hepatotoxicity, but their impact on clinical outcomes remains unclear [54].

### 5.12. Ayurvedic Herbal Products

The majority of India's population uses Ayurvedic herbal products with hepatotoxicity rarely reported. Ayurvedic herbal products include a wide variety of herbs including but not limited to Ashwagandha, Brahmi, Turmeric, Aloe Vera, Indian Mulberry, Indian Senna, and Malabar Tamarind—all of which have different liver safety profiles [56]. Contents and purity vary with 20–22% of US and Indian-manufactured Ayurvedic products purchased over the Internet found to be contaminated with lead, mercury, or arsenic [15]. In a 2019 single-center study from India, analysis of HDS (of which the most common were traditional Ayurvedic-polyherbal formulations) revealed the presence of adulterants such as heavy metals, NSAIDs, chemotherapy agents, anti-depressants, anti-biotics and more [57].

Taken for a multitude of reasons including strength, vigor, memory, anxiety, and fatigue, Ashwagandha is widely used in Southeast Asia and in recent years has been gaining increasing popularity in Western countries. Though no reports of elevated liver enzymes have been noted in clinical trials [58], a recent case series published in 2021 identified 5 cases of DILI attributed to the use of Ashwagandha. All patients developed jaundice with a peak total bilirubin ranging from 8.0–14.4 mg/dL within 2–12 weeks of taking their ashwagandha-containing supplement. Other reported symptoms included nausea, abdominal pain, fatigue, and pruritus. Patterns of liver injury were either cholestatic or mixed in nature and generally normalized 1–5 months after drug discontinuation. In the one case that was biopsied, a liver biopsy showed acute cholestatic hepatitis with prominent canalicular cholestasis without lobular inflammation [59]. It is important to consider the possibilities of drug mislabeling and drug adulteration but given the rising cases of reported liver injury from Ashwagandha, it is important for healthcare providers to be aware of Ashwagandha's risk for harm.

*5.13. Traditional Chinese Medicine*

More than 13,000 herbal preparations have been used for thousands of years as part of Traditional Chinese Medicine yet only a few (less than 60) have been implicated in liver injury [59]. Most products blend 4–5 herbs with 1–2 "King herbs" as the principal ingredient. Labels may only list the "King herb" and additional herbs may alter the constituents making the determination of causative compounds difficult [15]. Plant misidentification, species variation, differing harvest conditions, and extraction process may all contribute to inconsistent herbal products whereas clinical misuse and idiosyncratic reactions may result in hepatoxicity. Herb-induced hepatotoxicity accounts for 20–40% of drug-induced liver injury in China. The most reported are Lu Cha (*Camellia sinensis*) and mixtures containing *Camellia sinensis* including Chaso and Onshido. Other commonly implicated mixtures include the skullcap (*Scutellaria baicalensis*) containing Xiao Chai Hu Tang used for liver disease and the anti-aging and general tonic Shou Wu Pian (*Polygonum multiflorum*) [60].

## 6. Practical Approach to Drug and HDS Liver Injury for the Clinician

Drug-induced liver injury can be a challenging diagnosis given the wide variety of clinical presentations. Symptoms can often be vague and non-specific such as fatigue, pruritus, weakness, abdominal pain, jaundice, dark-colored urine, rash, or fever. The pattern (hepatocellular, cholestatic, or mixed) and degree of injury can also vary greatly depending on the offending agent that was ingested, and the quantity involved. The clinical picture of herbal and dietary supplement hepatotoxicity can be even more murky due to varying product information available and patient recall bias concerning supplement use. The United States Pharmacopeia-National Formulary (USP-NF) is the official authority for all prescription and over-the-counter medications and supplements and in addition to providing regulatory oversight, also provides a comprehensive compendium of products [61]. However, most cases of liver injury from herbal and dietary supplements are idiosyncratic in nature—the pattern of liver injury can be unpredictable and is thought to be dependent on host factors such as specific human leukocyte antigen types, environmental factors, and the active ingredient(s) involved [62,63].

Consequently, in patients with vague non-specific symptoms without an obvious etiology, basic labs including a hepatic panel should be acquired. Any degree of elevation in liver enzymes should prompt a thorough medication history including any recent new medications, antibiotics, herbal or over-the-counter supplements that have been started within the past 6 months. These drugs can be cross-referenced with LiverTox®—a free database maintained by the National Institute of Diabetes and Digestive and Kidney Diseases—which reviews the potential hepatoxicity of over 1000 medications and herbal supplements. Any non-essential medications should be stopped, and liver enzymes should be trended. Note depending on the drug involved, resolution of underlying injury can take up to 1–2 months or longer [13].

Most cases of DILI can be managed and monitored in the outpatient setting and referral to a liver specialist can be considered if a patient presents with liver enzymes >3–5× the upper limit of normal or if there is evidence of liver dysfunction (elevated INR, decreased albumin). However, in patients with severe symptoms such as nausea/vomiting, jaundice, or decompensated liver disease, admission should be considered for symptom management, expedited work-up of the patient's underlying disease, and consideration for liver biopsy. Notably, in patients who present with coagulopathy and altered mental status, urgent admission and subsequent transfer to a liver transplant center should be considered given these patients are at high risk for mortality once signs/symptoms of acute liver failure develop [62]. While there are few studies consistently linking patient risk factors for HDS liver injury, there are well-described risk factors for certain drug-induced hepatoxicity including age and female gender. However, underlying chronic liver disease does not appear to predispose one to DILI but may be associated with worse outcomes if one does not have enough reserve to recover [64].

Liver injury associated with HDS appears to be increasing as use becomes more prevalent. It is imperative to recognize the signs and symptoms of liver injury related to medications and HDS with potential causative agents held. The products reviewed above represent only a fraction of HDS linked to liver injury with additional offenders continuing to be identified. Suspected adverse events related to medications and HDS can be reported to the FDA's MedWatch system (http://www.fda.gov/Safety/MedWatch/default.htm (accessed on 4 September 2023)). The National Institute of Diabetes and Digestive and Kidney Diseases (NIDDK) and the National Institutes of Health have created LIVERTOX®, a current, searchable database of medications, herbs, and dietary supplements associated with hepatotoxicity at http://livertox.nih.gov/index.html (accessed on 4 September 2023).

## 7. Future Directions

Contrary to popular belief, HDS are no safer than conventional pharmaceuticals and many have been associated with severe liver injury. Supplements marketed for body-building and weight loss containing a multitude of ingredients are especially problematic. Product variability, contamination, and adulteration of HDS remain a concern under current regulation. Coupled with the idiosyncratic nature of HDS-induced liver injury, identifying the causative constituents in this environment is challenging. To guide future research, the National Institutes of Health and the American Association for the Study of Liver Diseases sponsored a Workshop on Liver Injury from Herbal and Dietary Supplements in May 2015. The most singular finding was that many botanical supplements are generally safe with appropriate selection and judicious use and that the cases of HDS-induced liver injury appear to contain adulterants or highly concentrated constituents, far beyond levels found in the plant or standard decoction. Appropriate adjudication and causality assignment are critical to identifying and classifying HDS hepatoxicity cases. Advancements in analyzing the chemical profiles of these products are paramount for future studies to elucidate mechanisms of toxicity and possible susceptibility genes. Meanwhile, physicians and scientists should continue to educate the public on the hepatotoxic potential of HDS.

**Author Contributions:** Conceptualization, J.K.-S.L. and S.R.T.; writing—original draft preparation, S.R.T.; writing—review and editing, J.K.-S.L. and S.R.T.; supervision, S.R.T. All authors have read and agreed to the published version of the manuscript.

**Funding:** This research received no external funding.

**Institutional Review Board Statement:** Not applicable.

**Informed Consent Statement:** Not applicable.

**Data Availability Statement:** Not applicable.

**Conflicts of Interest:** The authors declare no conflict of interest.

## Abbreviations

| | |
|---|---|
| ALT | Alanine transaminase |
| Alk P, AP | Alkaline phosphatase |
| CAM | Complementary and alternative medicine |
| CIOMS | Council for International Organizations of Medical Sciences |
| DILI | Drug-induced liver injury |
| DILIN | Drug Induced Liver Injury Network |
| EGCG | Epigallocatechin gallate |
| FDA | Food and Drug Administration |
| HDS | Herbal and dietary supplements |
| RUCLAM | Roussel Uclaf Causality Assessment Method |
| ULN | Upper limit of normal |
| Tbili | Total Bilirubin |

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
