# Peer review of "Hidden Dangers: Herbal and Dietary Supplement Induced Hepatotoxicity"

_livers, doi:10.3390/livers3040041_

Round 1

Reviewer 1 Report

Comments and Suggestions for Authors

Peer review report on “Hidden Dangers: Herbal and Dietary Supplement Induced Hepatotoxicity.”
Manuscript ID: livers-2642786

This paper reviews the knowledge available on the use and regulatory status of herbal and dietary supplements regarding their hepatoxicity and presents some of the most common supplements implicated in adverse events. It is well written and informative. However, while acknowledging the difficulty inherent in cataloguing and considering all the liver remedies available, it does suffer from a lack of depth when confronted with the problem of how many supplements to discuss while not overwhelming the paper with endless and tedious lists. If the authors can further moderate the paper to include more information as to what the consumer can do to avoid harm, it would amend somewhat the unavoidable conclusion the reader will reach that all herbal supplements are dangerous. It would thus also be more balanced. One trove of information which has only been mentioned in passing is the United States Pharmacopeia-National Formulary (USP-NF) which is the official reference for herbal preparations regarding quality standards. Please add this to your section on resources for the clinician.

Table 1 and 3 are blurry and difficult to read. Figure 2 is impossible to read. Please rectify.

Please check the fonts on the sub-headings for consistency.

Author Response

Thank you for taking the time to review our manuscript. Please see edits in uploaded document including changing of headings for consistency and additional section outlining practical approach for clinicians including USP reference.  Updated tables and figures for better readability uploaded. The RUCAM figure can also be found at Roussel Uclaf Causality Assessment Method (RUCAM) in Drug Induced Liver Injury - LiverTox - NCBI Bookshelf (nih.gov)

Reviewer 2 Report

Comments and Suggestions for Authors

Lin and Tujios conducted a delicate review work on the hidden dangers of herbal and dietary supplements and their potential to cause hepatotoxicity. Indeed, dietary supplements are not regulated by the FDA as strictly as drugs, and they still could cause adverse effects, just like any drugs due for toxicity studies. The authors emphasized the need for increased awareness of the potential risks associated with herbal and dietary supplements. The reviewer does have a few questions if the authors could address them.

Major

1)    It is a bit odd to start with an introduction to herbal medicine history, given that this article discusses the hepatotoxicity of herbal medicine or supplements. It would be interesting, however, if the authors could include the history of the toxicity of herbal medicine. Is there any ancient record regarding how these herbal regimens, although treating certain diseases, led to toxicity? This could further support the author's point. 

2)    Could the authors elaborate more on the patient's journey in the process of diagnosing herbal and supplement-induced liver injury? Since patients will be asymptomatic at the early stage of liver injury, are they mostly diagnosed during routine physical examinations or while examining for other conditions? The authors mentioned an algorithm for assessing HILI, but who are the HCPs that make the diagnosis? What will be the stage when they are diagnosed if symptomatic? Do they have hepatic manifestation that requires them to be hospitalized?

3)    The author should analyze and discuss the susceptible groups because the response to herbal and supplement-induced liver injury is idiosyncratic. For example, patients with underlying diseases such as NAFLD or NASH, hepatitis, or even chronic renal disease could potentially be more susceptible to herbal and supplement-induced liver injury. Are there subgroup analyses regarding this?

Minor

1)    Table 1, Table 3, and Figure 2 are screenshots with very low resolution. Please redo them in tables and increase the resolution for all figures.

2)    Figures and tables are not referred in the text. Please call out figures and tables in the text where they are used to support the authors’ argument. Otherwise, what is the point of having them in this article?

Author Response

Thank you for your insightful comments and questions. 

  1. Unfortunately true incidence of HDS liver injury is unknown but appears to be increasing over recent years. History of medicinal herbs provided as a backdrop to illustrate the increase in toxicity in the setting of limited  oversight and change in preparation leading to contamination and highly toxic chemicals. A sentence added to introduction to better emphasize this point.
  2. A practical approach to HDS liver injury for the clinician added to manuscript to better clarify the patient's presentation and management. 
  3. There are few papers identifying specific risk factors for susceptibility to certain herbs. However, there are generally accepted risk factors for DILI and few well described genetic susceptibilities to particular drugs. Underlying liver disease does not seem to increase risk to HDS liver injury but may be associated to poorer outcomes given decreased hepatic reserve. Sentences and reference added to point. 
  4. Updated figures uploaded for better resolution. 
  5. Tables and figures referenced in text. 
